# Influence of the Computer-Aided Static Navigation Technique and Mixed Reality Technology on the Accuracy of the Orthodontic Micro-Screws Placement. An In Vitro Study

**DOI:** 10.3390/jpm11100964

**Published:** 2021-09-27

**Authors:** Elena Riad Deglow, Sergio Toledano Gil, Álvaro Zubizarreta-Macho, María Bufalá Pérez, Paulina Rodríguez Torres, Georgia Tzironi, Alberto Albaladejo Martínez, Antonio López Román, Sofía Hernández Montero

**Affiliations:** 1Department of Implant Surgery, Faculty of Health Sciences, Alfonso X el Sabio University, 28691 Madrid, Spain; elenariaddeglow@gmail.com (E.R.D.); stolegii@myuax.com (S.T.G.); mperebuf@uax.es (M.B.P.); prodrtor@uax.es (P.R.T.); salvator@uax.es (A.L.R.); shernmon@uax.es (S.H.M.); 2Department of Orthodontics, Faculty of Medicine and Dentistry, University of Salamanca, 37008 Salamanca, Spain; georgiatzironi@usal.es (G.T.); albertoalbaladejo@usal.es (A.A.M.)

**Keywords:** orthodontics, micro-screws, orthodontic anchorage, mini-implants, temporary anchorage devices

## Abstract

To analyze the effect of a computer-aided static navigation technique and mixed reality technology on the accuracy of orthodontic micro-screw placement. **Material and methods**: Two hundred and seven orthodontic micro-screws were placed using either a computer-aided static navigation technique (NAV), a mixed reality device (MR), or a conventional freehand technique (FHT). Accuracy across different dental sectors was also analyzed. CBCT and intraoral scans were taken both prior to and following orthodontic micro-screw placement. The deviation angle and horizontal deviation were then analyzed; these measurements were taken at the coronal entry point and apical endpoint between the planned and performed orthodontic micro-screws. In addition, any complications resulting from micro-screw placement, such as spot perforations, were also analyzed across all dental sectors. **Results**: The statistical analysis showed significant differences between study groups with regard to the coronal entry-point (*p* < 0.001). The NAV study group showed statistically significant differences from the FHT (*p* < 0.001) and MR study groups (*p* < 0.001) at the apical end-point (*p* < 0.001), and the FHT group found significant differences from the angular deviations of the NAV (*p* < 0.001) and MR study groups deviations (*p* = 0.0011). Different dental sectors also differed significantly. (*p* < 0.001) Additionally, twelve root perforations were observed in the FHT group, while there were no root perforations in the NAV group. **Conclusions**: Computer-aided static navigation technique enable more accurate orthodontic micro-screw placement and fewer intraoperative complications when compared with the mixed reality technology and conventional freehand techniques.

## 1. Introduction

Anchorage systems pose a consistent issue in orthodontic treatments, as they are often uncomfortable, unattractive and their success relies heavily on patient cooperation [1]. The introduction of temporary anchorage devices (TAD) has drastically changed clinical treatment as they facilitate orthodontic treatments offering an alternative to conventional orthodontic treatments [2]. Currently, there are several anchored devices available for orthodontic purposes with the orthodontic micro-screws being the most popular due their small size characterized with smooth surfaces, which allow the orthodontic micro-screws to be loaded immediately after their insertion, as well as to cause less post-operative pain and to be removed easily after treatment [3]. The anchorage can be classified according to the location in intra-oral, extra-oral, or muscular; additionally, the anchorage can be classified in simple, stationary or reciprocal, according to the applied force and even in single, compound, multiple and demands or minimum, moderate, maximum and absolute depending on the anchorage units [4]. Furthermore, temporary skeletal anchorage devices have been successfully used to provide intra-oral absolute anchorage [5]. However, success rate and intra-operative complications related to orthodontic micro-screws can be affected by a number of variables, including the inherent characteristics attributed to the patient (age, gender, systematic diseases, periodontal status, smoking, skeletal pattern) [4], experience of the clinician [5], mechanical properties of the orthodontic micro-screw [6], patient cares [7], placement torque [8], placement site [9,10], cortical bone thickness [8,9], insertion angle [11], root proximity [12], bone density [13], bone stress [14], and orthodontic force [12,13]. Moreover, root contact is considered one of the main drawbacks related to orthodontic micro-screws placement that it is possible to occur during insertion [7,10,12]. Therefore, some approaches have been proposed based on cone-beam computed tomography (CBCT) scan [12], standard two-dimensional radiography [15], and panoramic radiography [16] to plan pre-operatively the insertion site of orthodontic micro-screws preventing root contact. Various insertion sites have been suggested according to the bone quality and low risk of root contact, such as edentulous areas, the palate and the zygomatic crest [17]; however, in most cases, the orthodontic micro-screws are inserted between the roots of contiguous teeth [17,18]. Unfortunately, complications derived from the orthodontic micro-screws are related to an incorrect insertion positioning which may lead to the trauma of the periodontal ligament [7,12,18], artery or nerve injury and even maxillary sinus perforation [19]. In addition, potential root damage by orthodontic micro-screw placement has been linked to severe side-effects, such as ankylosis, osteosclerosis, and loss of tooth vitality [7,10,12,18,20]. Therefore, it is mandatory to conduct an accurate pre-operative planning of orthodontic micro-screws placement site before the insertion procedure [21]. Consequently, a customized 3D-printed splint can be fabricated to provide a complete guide for the placement of orthodontic micro-screws [22]. In addition, augmented reality devices have been applied to improve the visualization [23] and experimentally improve the accuracy of conventional-length dental implant placement [24]; however, little literature has been published and clinical trials are necessary to assess the accuracy of this technology. Moreover, mixed reality appliances have not been previously used in the field of orthodontics and especially for orthodontic micro-screw placement and could be useful due to the accurate tracking technology.

The present study aims to analyze and evaluate the accuracy of orthodontic micro-screws and root contact prevalence, comparing a conventional freehand technique and a computer-aided static navigation technique in all dental sectors. The null hypothesis (H_0_) states that there is no difference in the accuracy of orthodontic micro-screw placement between a conventional freehand technique and a computer-aided static navigation technique at the coronal entry-point, apical end-point and angular deviation in all dental sectors.

## 2. Materials and Methods

### 2.1. Study Design

Upper teeth from all dental sectors, which required extraction due to periodontal and orthodontic reasons, were selected for study from cases treated at the Dental Centre of Innovation and Advanced Specialties at Alfonso X El Sabio University (Madrid, Spain), between February and April 2021. A randomized controlled in vitro study was carried out in compliance with the principles outlined by the German Ethics Committee’s statement on using organic tissues for medical research (Zentrale Ethikkommission, 2003). The study was authorized in November 2020 by the Ethical Committee of the Faculty of Health Sciences, Alfonso X el Sabio University (Madrid, Spain), in July 2021 (Process No. 21/2021). All patients gave their informed consent for their teeth to be used in the study.

### 2.2. Experimental Procedure

The teeth were placed in fourteen experimental models of epoxy resin (Ref. 20-8130-128, EpoxiCure^®^, Buehler, IL, USA) with 16 teeth each. A silicone splint was created by a conventional impression to a dental training model of acrylic resin, and the teeth were placed on it. Subsequently, the epoxy resin (Ref. 20-8130-128, EpoxiCure^®^, Buehler, IL, USA) was mixed following the manufacturer’s recommendations and poured inside the silicone splint with the teeth. After the epoxy resin setting the silicone splint was removed from the epoxy resin model. A bilateral Student’s *t*-test of two independent samples was used to achieve a power of 80.00% for assessing differences from the null hypothesis H_0_: μ_1_ = μ_2_, taking into account that the significance level is 5.00%, it will be necessary to include 207 orthodontic micro-screws. The orthodontic micro-screws (Dual Top^®^ Anchor System, JEIL Medical Corporation, Guro-gu, Seoul, Korea) were randomly assigned (Epidat 4.1, Galicia, Spain) to one of the following study groups: Group A: Orthodontic micro-screws placement in the incisive-canine sector by a computer-aided static navigation technique (NemoScan^®^, Nemotec, Madrid, Spain) (NAV-i) (*n* = 23), B: Orthodontic micro-screws placement in the incisive-canine sector by a mixed reality device (Hololens1, Redmond, WA, USA) (MR-i) (*n* = 23), C: Orthodontic micro-screws placement in the incisive-canine sector by conventional freehand technique (FHT-i) (*n* = 23), Group D: Orthodontic micro-screws placement in the premolar sector by a computer-aided static navigation technique (NemoScan^®^, Nemotec, Madrid, Spain) (NAV-p) (*n* = 23), E: Orthodontic micro-screws placement in the premolar sector by a mixed reality device (Hololens1, Redmond, WA, USA) (MR-p) (*n* = 23), F: Orthodontic micro-screws placement in the premolar sector by conventional freehand technique (FHT-p) (*n* = 23), Group G: Orthodontic micro-screws placement in the molar sector by a computer-aided static navigation technique (NemoScan^®^, Nemotec, Madrid, Spain) (NAV-m) (*n* = 23), H: Orthodontic micro-screws placement in the molar sector by a mixed reality device (Hololens1, Redmond, WA, USA) (MR-m) (*n* = 23) and Group I: Orthodontic micro-screws placement in the molar sector by conventional freehand technique (FHT-m) (*n* = 23). The teeth assigned to both experimental models presented similar anatomical dimensions evaluated with an electronic caliper and were positioned in the experimental model using a silicone splint to prevent different interradicular spaces between the different teeth of the experimental models.

A preoperative cone-beam computed tomography (CBCT) scan (WhiteFox, Acteón Médico-Dental Ibérica S.A.U.-Satelec, Merignac, France) was taken of the experimental epoxy resin models (Ref. 20-8130-128, EpoxiCure^®^, Buehler, IL, USA) using the following exposure parameters: 105.0 kV peak, 8.0 milliamperes, 7.20 s, and a field of view of 15 × 13 mm (Figure 1A). A 3D surface scan was subsequently performed via 3D intraoral scan (True Definition, 3M ESPE ™, Saint Paul, MN, USA) using three-dimensional in-motion video imaging technology (Figure 1B). The datasets obtained from the digital workflow were added to 3D implant planning software (NemoScan^®^, Nemotec, Madrid, Spain) in order to plan the virtual placement of the orthodontic micro-screws (Ref. 16-G2-008, Dual Top^®^ Anchor System, JEIL Medical Corporation, Guro-gu, Seoul, Korea). The screws were 1.3mm in diameter, 8.0mm in length in the active part and 2.0mm in the inactive part. Virtual placement was planned by matching the three-dimensional surface scan with data from the CBCT, with the key points being overlaid on the crown of the teeth (Figure 1C). Virtual orthodontic micro-screws were placed to a depth of 6mm, an insertion angle of 90° to the longitudinal axis of the teeth, and a depth of 6.0mm with respect to the cortical plate (Figure 1D).

The orthodontic micro-screw placement of the experimental model randomly sorted into the NAV study group were virtually planned on the 3D implant planning software (NemoScan^®^, Nemotec, Madrid, Spain). Afterwards, the surgical template was designed (Figure 1E) and manufactured (NemoScan^®^, Nemotec, Madrid, Spain) by 3D-printed techniques (Figure 1F). The interradicular spaces where the orthodontic micro-screws were placed were also randomly selected (Epidat 4.1, Galicia, Spain).

The orthodontic micro-screws (Dual Top^®^ Anchor System, JEIL Medical Corporation, Guro-gu, Seoul, Korea) randomly assigned to the MR study group were virtually planned using the 3D implant-planning software (NemoScan, Nemotec, Madrid, Spain) with the measures previously described. The STL digital file of the dental implants positioning was uploaded to a mixed reality appliance (Hololens1, Redmond, WA, USA), to allow the orthodontic micro-screws (Dual Top^®^ Anchor System, JEIL Medical Corporation, Guro-gu, Seoul, Korea) placement procedure in all space planes (INNOAREA, Valencia, Spain) (Figure 2A–F).

The orthodontic micro-screws (Dual Top^®^ Anchor System, JEIL Medical Corporation, Guro-gu, Seoul, Korea) randomly assigned to the FHT study group were placed in the experimental models by a unique operator per group, according to the recommendations performed by Cozzani et al. [25] to place self-tapping orthodontic micro-screws after using an osteotomy pilot drill (Ref.: 112-MC.201, Dual Top^®^ Anchor System, JEIL Medical Corporation, Guro-gu, Seoul, Korea). All orthodontic micro-screws (Dual Top^®^ Anchor System, JEIL Medical Corporation, Guro-gu, Seoul, Korea) of the NAV, MR and FHT study groups were inserted in the middle of the inter-root space, 2 mm from the alveolar ridge.

### 2.3. Measurement Procedure

After placing the orthodontic micro-screws (Dual Top^®^ Anchor System, JEIL Medical Corporation, Guro-gu, Seoul, Korea), postoperative CBCT scans were taken of the experimental models. Virtual orthodontic micro-screw (Dual Top^®^ Anchor System, JEIL Medical Corporation, Guro-gu, Seoul, Korea) pre- and post-operative CBCT scans of the study groups were added to the 3D implant planning software (NemoScan^®^, Nemotec, Madrid, Spain). These images were then matched to assess the deviation angle (as measured in the middle of the cylinder) and horizontal deviation (taken at the coronal entry-point and apical end-point) (Figure 3A–D) by an independent observer.

Root perforations arising from the placement of the orthodontic micro-screws (Dual Top^®^ Anchor System, JEIL Medical Corporation, Guro-gu, Seoul, Korea) placement were also analyzed and recorded at the 3D implant planning software (NemoScan^®^, Nemotec, Madrid, Spain) between the conventional freehand technique, mixed reality technique and computer-aided static navigation technique (Figure 4A–D).

### 2.4. Statistical Tests

All studied variables were recorded using SPSS 22.00 for Windows for statistical analysis. The descriptive statistical analysis used the mean and standard deviation (SD) of quantitative variables. A multivariate (generalized linear model (GLM)) was used for analyzing the effect of the study group, the dental group and the interaction between the variables in each of the response variables. In case of obtaining a significant result, 2 to 2 tests were carried out a posteriori. To correct the type I error, the *p*-values were corrected using the Tukey correction. As the variables were normally distributed; *p* < 0.05 was determined statistically significant.

## 3. Results

The means and SD values for the coronal entry-point, apical end-point and angular deviation of the computer-aided static navigation technique, mixed reality technique and conventional freehand technique orthodontic micro-screws in all dental sectors are displayed in Table 1.

Statistically significant differences were shown between the computer-aided static navigation technique and conventional freehand technique study groups (*p* < 0.001), the computer-aided static navigation technique and the mixed reality technique study groups (*p* < 0.001) and the mixed reality technique and conventional freehand technique study groups with regard to the coronal entry point deviations of planned and placed orthodontic micro-screws (*p* < 0.001) (Figure 5).

In addition, the means and SD values for coronal entry-point deviations of the computer-aided static navigation technique, mixed reality technique and conventional freehand technique orthodontic micro-screws in the incisive-canine, premolar and molar dental sector are displayed in Table 2.

Statistically significant differences were also shown between the coronal entry-point deviations of the orthodontic micro-screws placed using the computer-aided static navigation technique and conventional freehand technique in the incisive-canine premolar and molar dental sectors (*p* < 0.001). Moreover, statistically significant differences were also shown between the coronal entry-point deviations of the orthodontic micro-screws placed using the mixed reality technique and conventional freehand technique in the incisive-canine and molar dental sectors (*p* < 0.001); however, no statistical significances were shown at the premolar dental sector (*p* = 0.7013). Finally, statistically significant differences were also shown between the coronal entry-point deviations of the orthodontic micro-screws placed using the mixed reality technique and computer-aided static navigation technique in the incisive-canine and premolar dental sectors (*p* < 0.001); however, no statistical significances were shown at the molar dental sector (*p* = 0.1901) (Figure 6).

Specifically, the means and SD values for coronal entry-point deviations of the orthodontic micro-screws in the selected tooth positioning are displayed in Table 3 and Figure 6.

Statistically significant differences were also shown between the coronal entry-point deviations of the orthodontic micro-screws placed in the incisive-canine and premolar dental sectors (*p* = 0.0115), incisive-canine and molar dental sectors (*p* < 0.001) and premolar and molar dental sectors (*p* < 0.001) (Figure 7).

Additionally, statistically significant differences at the apical end-point deviations of planned and placed orthodontic micro-screws between the computer-aided static navigation technique and conventional freehand technique study groups (*p* < 0.001) and the computer-aided static navigation technique and the mixed reality technique study groups (*p* < 0.001). However, no statistical significant differences were shown between the mixed reality technique and conventional freehand technique study groups (*p* = 0.0598) (Figure 8).

In addition, the means and SD values for apical end-point deviation of the computer-aided static navigation technique, mixed reality technique and conventional freehand technique orthodontic micro-screws in the incisive-canine, premolar and molar dental sector are displayed in Table 4.

Statistically significant differences were also shown between the apical end-point deviations of the orthodontic micro-screws placed using the computer-aided static navigation technique, mixed reality technique and conventional freehand technique in the incisive-canine premolar and molar dental sectors (*p* < 0.001), except for the comparison between the mixed reality technique and conventional freehand technique study groups at the premolar dental sector (*p* = 0.2837) and between the computer-aided static navigation technique and the mixed reality technique at the molar sector (*p* = 0.3097) (Figure 9).

Specifically, the means and SD values for apical end-point deviations of the orthodontic micro-screws in the selected tooth positioning are displayed in Table 5 and Figure 10.

Statistically significant differences were also shown between the apical end-point deviations of the orthodontic micro-screws placed in the incisive-canine and premolar dental sectors (*p* < 0.001), incisive-canine and molar dental sectors (*p* < 0.001) and premolar and molar dental sectors (*p* < 0.001).

Furthermore, statistically significant differences in the angular deviations of planned and placed orthodontic micro-screws between the computer-aided static navigation technique and conventional freehand technique study groups (*p* < 0.001), and between the mixed reality technique and conventional freehand technique study groups (*p* = 0.0011). However, no statistically significant differences were shown between the computer-aided static navigation technique and the mixed reality technique (*p* = 0.2603) (Figure 11).

In addition, the means and SD values of the angular deviation of the computer-aided static navigation technique, mixed reality technique and conventional freehand technique orthodontic micro-screws in the incisive-canine, premolar and molar dental sector are displayed in Table 6.

Statistically significant differences were not shown between the angular deviations of the dental groups variable (*p* = 0.8050); therefore, comparisons were not performed (Figure 12).

Specifically, the means and SD values for angular deviations of the orthodontic micro-screws in the selected tooth positioning are displayed in Table 7 and Figure 13.

Statistically significant differences were not shown between the angular deviations of the tooth positioning variable (*p* = 0.8050); therefore, comparisons were not performed (Figure 13).

Twelve root perforations were observed in the conventional freehand technique study group after the orthodontic micro-screws placement at teeth 1.2, 1.4, 1.6, 1.7, 1.8, 2.4, 2.5, 2.6, 2.7 and 2.8, which match with the highest coronal entry-point and apical end-point deviation values. No root perforations were identified in the computer-aided static navigation technique and mixed reality study groups.

## 4. Discussion

The results of the present study reject the null hypothesis (H_0_) that posits that there is no difference between the conventional freehand technique and computer-aided navigation technique at coronal entry-point, apical end-point and angular deviation, nor in the intraoperative complications.

The present study showed higher deviations for the conventional freehand technique than the computer-aided static navigation technique at the coronal entry point, apical end-point and angular values. Previous studies have analyzed the importance of surgical templates in the accuracy of orthodontic micro-screws placement [22,26,27,28,29]. Cassetta et al. also showed similar results and reported that the surgical template reduced considerably the coronal, apical and angular deviations for palatal micro-screws placement [30]. Moreover, Qiu et al. reported that the surgical templates used for orthodontic micro-screws placement provide safer and more stable micro-screws insertion than the conventional freehand technique [31]. In addition, Suzuki reported promising results related to the accuracy of orthodontic micro-screws placed by surgical template, although the results were analyzed using 2D-periapical radiographs [22]. Some insertion sites of orthodontic micro-screws have been recommended to prevent the damage of root processes, such as the zygomatic crest and mandibular buccal shelf area, although the most commonly used insertion sites are at the alveolar processes between dental roots [21]. Moreover, the orthodontic micro-screws usually can be inserted from the buccal side and they are commonly placed between the second premolar and first molar to ensure maximum anchorage [20]. The interdental space between the second premolar and first molar at 5 mm from the alveolar crest is generally about 3.0 mm [20]. This space might be insufficient for an orthodontic micro-screw with a diameter ranging from 1.2 mm to 2.0 mm. Even though root contact can be prevented by careful monitoring of the surgical procedure and the use of radiograph, CT, or surgical stent, the orthodontic micro-screw might be near enough to the root that it can histologically affect the root surface and surrounding tissues [22].

Orthodontic micro-screws have reported a mean failure rate of 13.5%, which is a modestly small rate, demonstrating their effectiveness in clinical practice [20]. Furthermore, one of the most common complications reported during orthodontic micro-screws insertion is causing a trauma to the dental root and/or the periodontal ligament; specifically, when the trauma is limited to the outer dental root surface without pulp involvement, it is less probably to influence the prognosis of the tooth [32]; in addition, the periodontal ligament and the cementum have shown a complete reparation capacity between 12 and 18 weeks after the orthodontic micro-screws removal [33]. Moreover, when orthodontic micro-screws insertion comprises the periodontal ligament, the patient begins to experience stronger sensations under local anesthesia [15,34]. Furthermore, if there is contact with the root, the orthodontic micro-screws may require greater insertion strength [33]. Finally, if the clinician suspects trauma to the tooth or periodontium, it is mandatory to unscrew immediately the orthodontic micro-screw 2 to 3 turns and assess the position radiographically [35]. In the present study, twelve orthodontic micro-screws placed by conventional freehand technique caused root perforation and no root contact was shown in the teeth randomly assigned to computer-aided static navigation technique. In addition, Kalra et al. analyzed the planning performed by CBCT scan and 2D-radiograph to prevent root perforations, and concluded that the planning performed by CBCT scan showed no root perforations and the planning performed by 2D-radiograph showed three root perforations in twenty patients [21]. Moreover, Bufalá Perez et al. analyzed the influence of clinician experience on the accuracy of placement of orthodontic self-tapping micro-screws and reported five out of thirty root perforations in the study group with no experience, compared with no root perforation in the group placed by an orthodontist with 10 years of experience [5].

The orthodontic micro-screw placement between contiguous roots necessitates proper radiographic planning, including a surgical template, as well as panoramic and periapical X-rays in order to determinate the safest placement site [16,36,37,38,39,40,41].

Severe bone damage from the insertion of orthodontic micro-screws can result in bone remodeling and trigger root resorption. Should the periodontal ligature be severely injured and if bone forms toward the reabsorbed root, the ligature is not able to protect the root and may lead to tooth ankylosis [42]. In addition, root resorption can be triggered by stimulating activation of the periodontal ligament in differentiatied cementoclasts. The incidence of root resorption can be limited if minimal injury is experienced during the orthodontic micro-screws insertion procedure. Seemingly heavy injury during insertion can cause root resorption even if there is no proximity of the orthodontic micro-screws and the root [33]. Motoyosi et al. categorizes the root proximity of orthodontic micro-screws into three groups: A: no contact between the root and orthodontic micro-screws, B: one point of contact between the root and the orthodontic micro-screws, and C: two or more points of contact [43]. Moreover, in the most severe cases causing loss of pulp vitality, ankylosis and root resorption are rare complications. Finally, the risk of pathology increases rapidly when orthodontic micro-screws are more proximal to the dental root surface, with critical proximity found to be 1mm. For those reasons it is important to predict the accurate position of the orthodontic micro-screws, because except for tissue damage, the contact of orthodontic micro-screws to the root may also provoke the loss of orthodontic micro-screws stability [44].

Augmented reality technology has been used by the education industry and manufacturers for improving dental education, especially in the field of dental implants [23]. In addition, augmented reality devices have been applied as well to visualize maxillectomy defects; however, a target image with a symbol track marker was necessary to superimpose the virtual models on the real scene by the augmented reality application [24]. This limitation has been solved in the present study by improving the tracking method without track markers. Furthermore, Ma et al. analyzed experimentally the accuracy of augmented reality technology for dental implant placement and reported a mean target error between 1.25 and 1.63 mm and a mean angle error between 4.03° and 6.10° [45]. Jiang et al. showed a mean horizontal deviation < 1.5 mm and a mean angular deviation < 1.5° [46]. However, mixed reality technology combining augmented and virtual reality still have not been used for dental implant placement. In the present study, the zygomatic dental implants placed by the mixed reality device showed higher coronal and angular deviations compared to the computer-aided static and dynamic study groups and free-hand control group; however, operator sensations were very promising and further studies are recommended.

The present study has the strength of including a sample size higher than the previous studies of Qiu et al. (*n* = 30) [27], Liu et al. (*n* = 34) [26], Miyazawa et al. (*n* = 44) [29] and Bae et al. (*n* = 45) [28], as well as presenting the results regarding the dental sector where the micro-screws were placed. This methodology aims to establish more reliable results as far as it concerns the morphology of the roots in those specific areas and the interdental distance which differs between each dental sector. On the other hand, it is an in vitro study with extracted teeth.

## 5. Conclusions

In conclusion, bearing in mind the limitations of this in vitro study, the results show that the computer-aided static navigation technique has an effect on the accuracy of orthodontic micro-screws placement and results in fewer intra-operative complications.

## Figures and Tables

**Figure 1 jpm-11-00964-f001:**
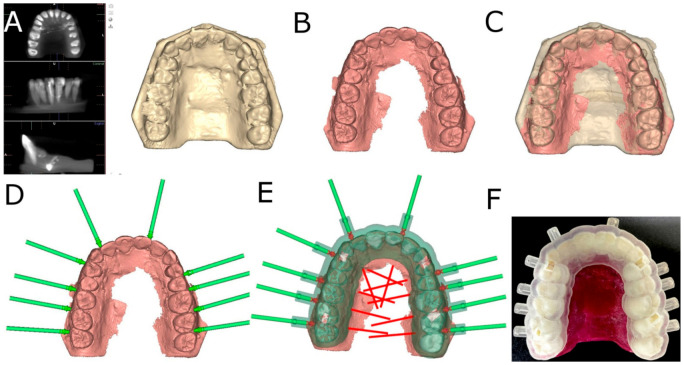
(**A**) DICOM files from the CBCT scan, (**B**) STL digital file from the digital impression, (**C**) alignment procedure between STL and CBCT scan digital files, (**D**) orthodontic micro-screws planning position, (**E**) surgical template design and (**F**) manufacturing.

**Figure 2 jpm-11-00964-f002:**
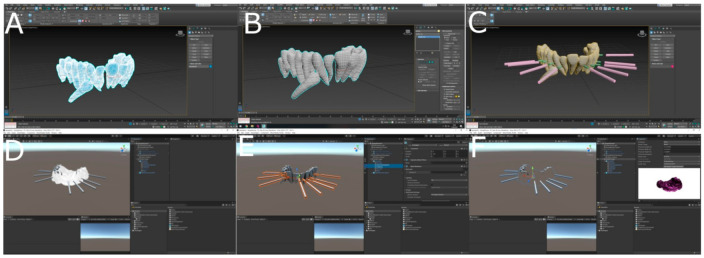
(**A**–**F**) Planning process in mixed reality device software.

**Figure 3 jpm-11-00964-f003:**
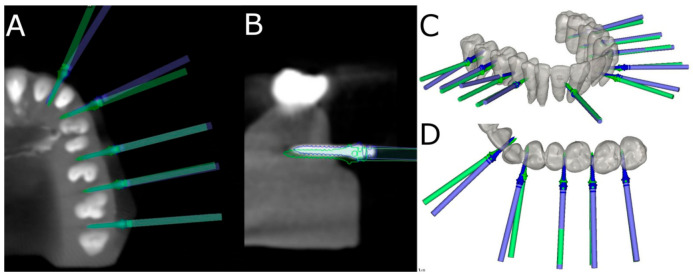
(**A**–**D**) Deviations measurement procedure between planned (green cylinder) and placed (blue cylinder) orthodontic micro-screws in the computer-aided static navigation technique study group.

**Figure 4 jpm-11-00964-f004:**
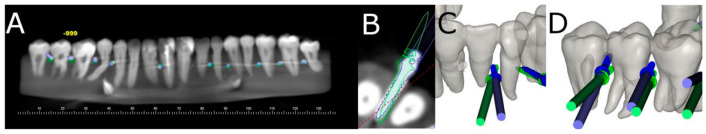
(**A**) Radiographic analysis of the root perforation in the 3D implant planning software, (**B**) Relationship between the root processes and the planned (green micro-screw) and performed (blue micro-screw) orthodontic micro-screws using a computer-aided static navigation technique (**C**) by the conventional free-hand technique and (**D**) mixed reality technique.

**Figure 5 jpm-11-00964-f005:**
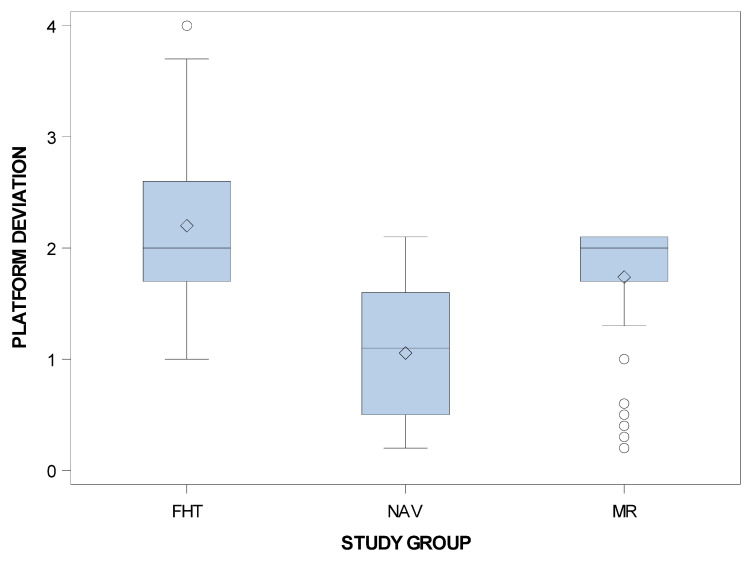
Box plot of the coronal deviations in planned and placed orthodontic micro-screws comparing the computer-aided static navigation technique, mixed reality technique and conventional freehand technique study groups.

**Figure 6 jpm-11-00964-f006:**
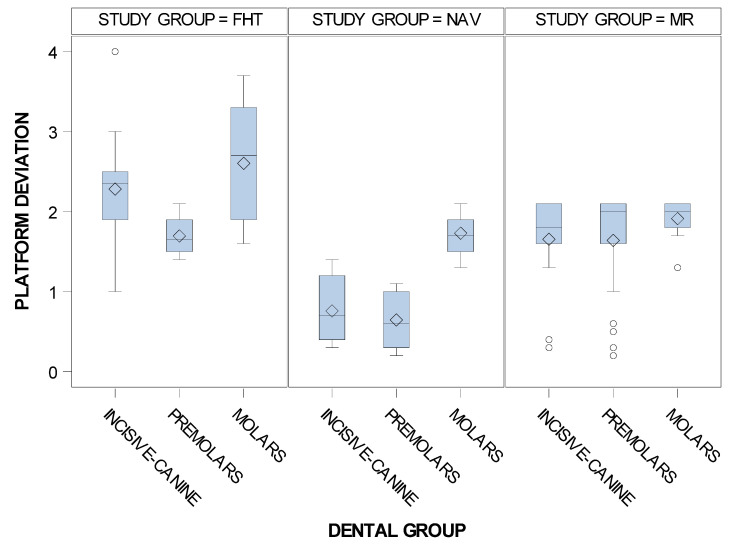
Box plot of the coronal entry-point deviations of the orthodontic micro-screws placed using the computer-aided static navigation technique, mixed reality technique and conventional freehand technique in the incisive-canine, premolar and molar dental sectors.

**Figure 7 jpm-11-00964-f007:**
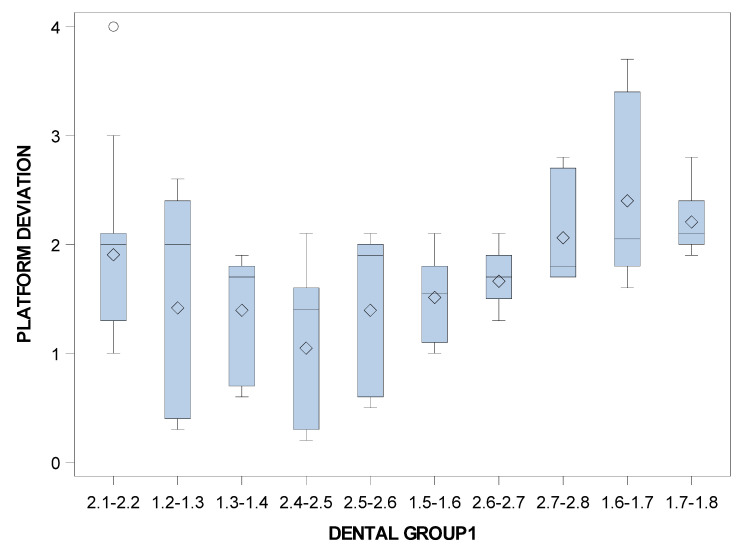
Box plot of the coronal entry-point deviations of the orthodontic micro-screws in the selected tooth positioning.

**Figure 8 jpm-11-00964-f008:**
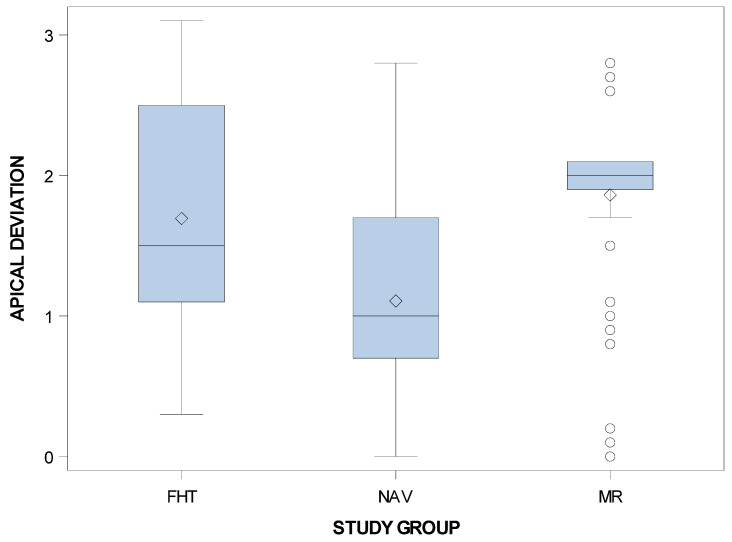
Box plot of the apical deviations in planned and placed orthodontic micro-screws between computer-aided static navigation technique, mixed reality technique and conventional freehand technique study groups.

**Figure 9 jpm-11-00964-f009:**
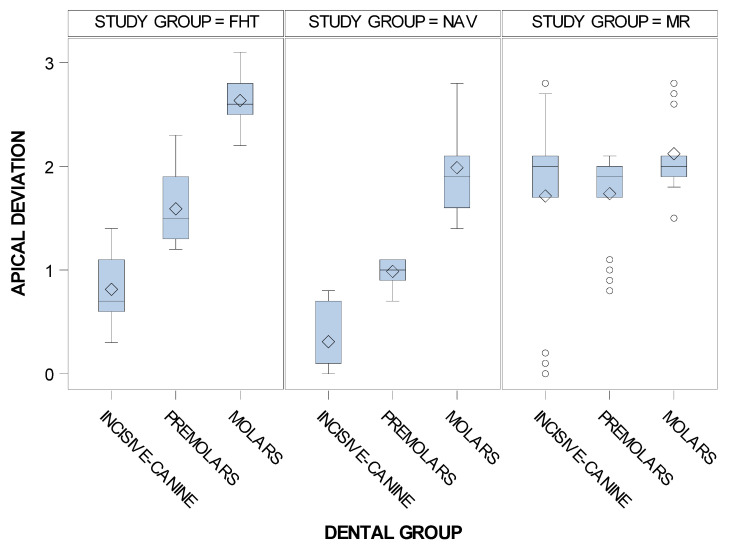
Box plot of the apical end-point deviations of the orthodontic micro-screws placed using the computer-aided static navigation technique, mixed reality technique and conventional freehand technique in the incisive-canine, premolar and molar dental sectors.

**Figure 10 jpm-11-00964-f010:**
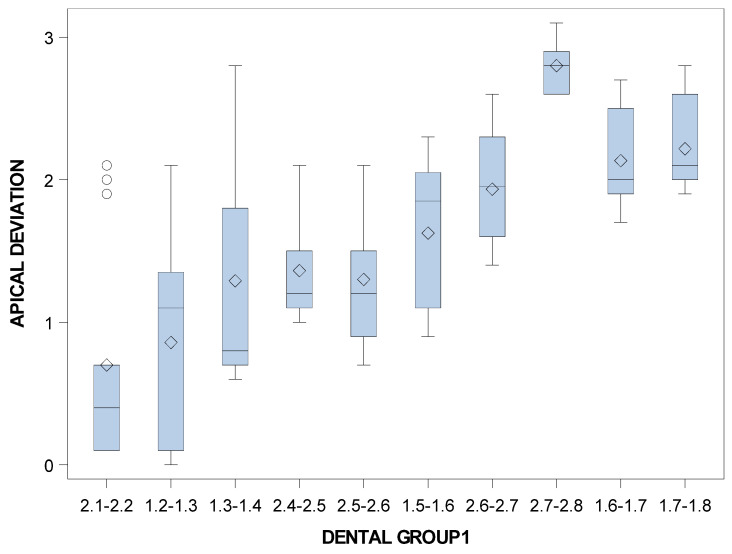
Box plot of the apical end-point deviations of the orthodontic micro-screws in the selected tooth positioning.

**Figure 11 jpm-11-00964-f011:**
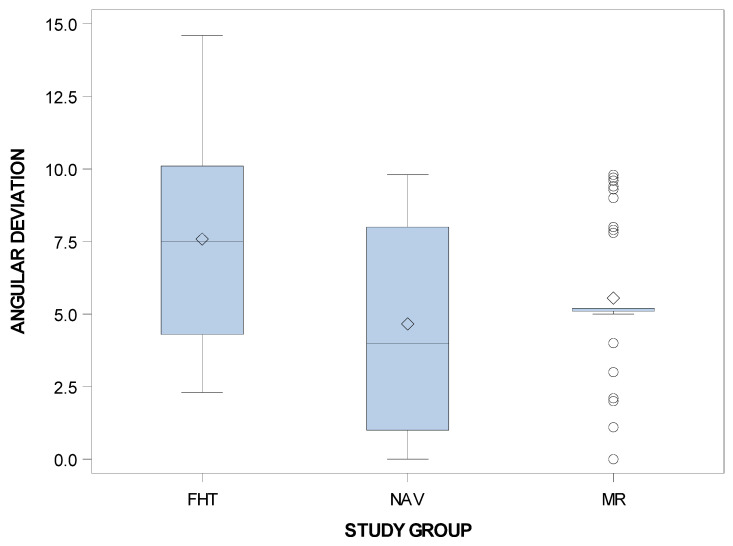
Box plot of angular deviations in planned and placed orthodontic micro-screws between the computer-aided static navigation technique, mixed reality technique and conventional freehand technique study groups.

**Figure 12 jpm-11-00964-f012:**
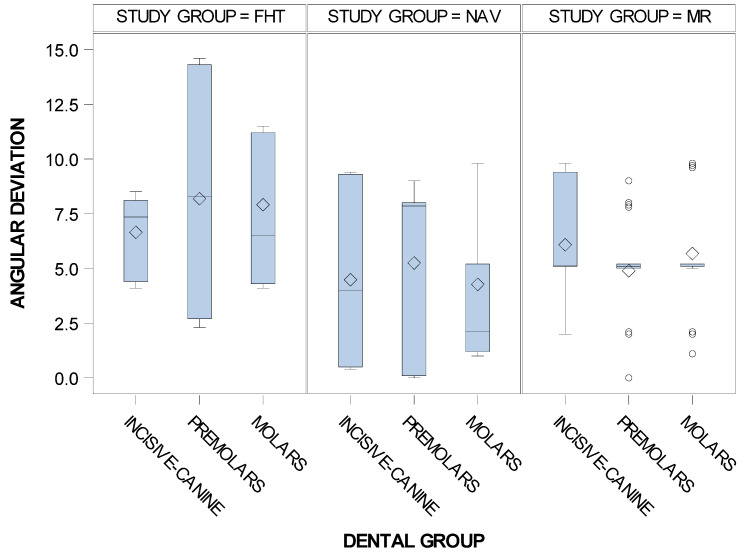
Box plot of the angular deviations of the orthodontic micro-screws placed using the computer-aided static navigation technique, mixed reality technique and conventional freehand technique in the incisive-canine, premolar and molar dental sectors.

**Figure 13 jpm-11-00964-f013:**
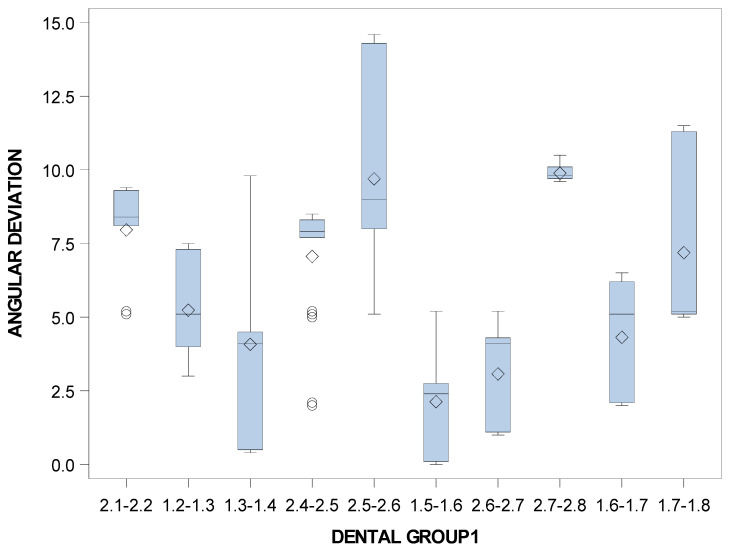
Box plot of the angular deviations of the orthodontic micro-screws in the selected tooth positioning.

**Table 1 jpm-11-00964-t001:** Descriptive deviation values at the coronal entry-point (mm), apical end-point (mm), and angular (°) levels of the orthodontic micro-screws placed by using conventional freehand technique, mixed reality technique and computer-aided static navigation technique study groups.

		*n*	Mean	SD	Minimum	Maximum
Coronal	NAV	69	1.06	0.59	0.20	2.10
	MR	69	1.74	0.52	0.20	2.10
	FHT	69	2.20	2.00	1.00	4.00
Apical	NAV	69	1.11	0.77	0.10	2.80
	MR	69	1.86	0.65	0.00	2.80
	FHT	69	1.69	0.82	0.40	3.10
Angular	NAV	69	4.66	3.65	0.00	9.80
	MR	69	5.55	2.46	0.00	9.80
	FHT	69	7.58	3.50	2.30	14.60

**Table 2 jpm-11-00964-t002:** Descriptive deviation values at coronal entry-point (mm) of the orthodontic micro-screws placed by using conventional freehand technique, mixed reality technique and computer-aided static navigation technique study groups in the incisive-canine, premolar and molar dental sector.

		*n*	Mean	SD	Minimum	Maximum
Inci sive-canine	NAV	23	0.76	0.39	0.30	1.40
	MR	23	1.65	0.57	0.30	2.10
	FHT	23	2.28	0.63	1.00	4.00
Premolar	NAV	23	0.65	0.35	0.20	1.10
	MR	23	1.64	0.65	0.20	2.10
	FHT	23	1.70	0.25	1.40	2.10
Molar	NAV	23	1.73	0.24	1.30	2.10
	MR	23	1.91	0.20	1.30	2.10
	FHT	23	2.60	0.65	1.60	3.70

**Table 3 jpm-11-00964-t003:** Descriptive deviation values at coronal entry-point (mm) of the orthodontic micro-screws placed in the selected tooth positioning.

	*n*	Mean	SD	Minimum	Maximum
2.1–2.2	14	1.90	0.78	1.00	4.00
1.2–1.3	14	1.42	0.99	0.30	2.60
1.3–1.4	13	1.40	0.53	0.60	1.90
2.4–2.5	14	1.05	0.73	0.20	2.10
2.5–2.6	14	1.40	0.75	0.50	2.10
1.5–1.6	14	1.51	0.40	1.00	2.10
2.6–2.7	13	1.66	0.25	1.30	2.10
2.7–2.8	14	2.06	0.47	1.70	2.80
1.6–1.7	14	2.40	0.78	1.60	3.70
1.7–1.8	14	2.21	0.28	1.90	2.80

**Table 4 jpm-11-00964-t004:** Descriptive deviation values at apical end-point (mm) of the orthodontic micro-screws placed by using conventional freehand technique, mixed reality technique and computer-aided static navigation technique study groups in the incisive-canine, premolar and molar dental sector.

		*n*	Mean	SD	Minimum	Maximum
Inci sive-canine	NAV	23	0.31	0.32	0.00	0.80
	MR	23	1.71	0.95	0.00	2.80
	FHT	23	0.81	0.34	0.30	1.40
Premolar	NAV	23	0.99	0.12	0.70	1.10
	MR	23	1.74	0.46	0.80	2.10
	FHT	23	1.59	0.35	1.20	2.30
Molar	NAV	23	1.99	0.43	1.40	2.80
	MR	23	2.12	0.34	1.50	2.80
	FHT	23	2.63	0.25	2.20	3.10

**Table 5 jpm-11-00964-t005:** Descriptive deviation values at apical end-point (mm) of the orthodontic micro-screws placed in the selected tooth positioning.

	*n*	Mean	SD	Minimum	Maximum
2.1–2.2	14	0.70	0.78	0.10	2.10
1.2–1.3	14	0.86	0.80	0.00	2.10
1.3–1.4	13	1.29	0.87	0.60	2.80
2.4–2.5	14	1.36	0.37	1.00	2.10
2.5–2.6	14	1.30	0.48	0.70	2.10
1.5–1.6	14	1.63	0.49	0.90	2.30
2.6–2.7	13	1.93	0.39	1.40	2.60
2.7–2.8	14	2.80	0.18	2.60	3.10
1.6–1.7	14	2.13	0.34	1.70	2.70
1.7–1.8	14	2.22	0.33	1.90	2.80

**Table 6 jpm-11-00964-t006:** Descriptive deviation values at angular level (°) of the orthodontic micro-screws placed by using conventional freehand technique, mixed reality technique and computer-aided static navigation technique study groups in the incisive-canine, premolar and molar dental sector.

		*n*	Mean	SD	Minimum	Maximum
Inci sive-canine	NAV	23	4.48	3.67	0.40	9.40
	MR	23	6.08	2.65	2.00	9.80
	FHT	23	6.65	1.72	4.10	8.50
Premolar	NAV	23	5.25	4.03	0.00	9.00
	MR	23	4.89	2.19	0.00	9.00
	FHT	23	8.18	4.98	2.30	14.60
Molar	NAV	23	4.27	3.32	1.00	9.80
	MR	23	5.68	2.47	1.10	9.80
	FHT	23	7.91	2.99	4.10	11.50

**Table 7 jpm-11-00964-t007:** Descriptive deviation values at angular level (°) of the orthodontic micro-screws placed in the selected tooth positioning.

	*n*	Mean	SD	Minimum	Maximum
2.1–2.2	14	7.96	1.69	5.10	9.40
1.2–1.3	14	5.24	1.67	3.00	7.50
1.3–1.4	13	4.08	3.60	0.40	9.80
2.4–2.5	14	7.06	1.98	2.00	8.50
2.5–2.6	14	9.70	3.70	5.10	14.60
1.5–1.6	14	2.13	1.93	0.00	5.20
2.6–2.7	13	3.07	1.76	1.00	5.20
2.7–2.8	14	9.89	0.28	9.60	10.50
1.6–1.7	14	4.32	1.91	2.00	6.50
1.7–1.8	14	7.19	3.02	5.00	11.50

## Data Availability

Information available on request in keeping with relevant restrictions (privacy or ethical).

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
