# Peer review of "Influence of the Computer-Aided Static Navigation Technique and Mixed Reality Technology on the Accuracy of the Orthodontic Micro-Screws Placement. An In Vitro Study"

_jpm, 2021, doi:10.3390/jpm11100964_

Round 1

Reviewer 1 Report

This is a very interesting paper and important in the field of orthodontics and the use of osseointegration to assist in tooth movement.  Although navigation has proven from your study to be more accurate, it would be interesting to consider guided surgery vs navigation, vs freehand to compare which is more accurate -guided surgery or navigation.

Author Response

Dear Reviewer 1,

I’m pleased to resubmit the manuscript of the work entitled, “Influence of the Computer-Aided Static Navigation Technique and Mixed Reality Technology on the Accuracy of the Orthodontic Micro-screws Placement. An in Vitro Study”

Reviewer 1: English language and style are fine/minor spell check required

Response: In order to adapt to the reviewer's 1 comments, we have send the manuscript to a specialized traductor.

Reviewer 1: This is a very interesting paper and important in the field of orthodontics and the use of osseointegration to assist in tooth movement.  Although navigation has proven from your study to be more accurate, it would be interesting to consider guided surgery vs navigation, vs freehand to compare which is more accurate -guided surgery or navigation.

Response: In order to adapt to the reviewer's 1 comments, we have improved the study by including another navigation study group using mixed reality technology.

We take this opportunity to thank the recommendations and suggestions made by the reviewers to improve the document.

Yours sincerely,

Reviewer 2 Report

The paper itself is well written and documented, showing a great effort from the authors. The topic sounds original and with an interesting clinical meaning and the large sample size is a point of strength as suggested by the authors. Placement of mini screw is now largely included in modern orthodontic treatments and investigating which placement system is more safely and accurate is an original topic.

Introduction is adequate and with recent references.

The groups described in the material and methos section are different than the ones described in the abstract section. Please correct the abstract section describing that the groups included were from A to F.

Is there any reference supporting the accuracy of the experimental models made by epoxy resin? If yes, please add as citation.

There are a lot of tables and plots. Maybe it could be better to unify some tables or plots.

Statistical analysis seems adequate and conclusions are correctly supported by the results.

Please correct the pagination from line 378.

Otherwise, the article is good.

Author Response

Dear Reviewer 2,

I’m pleased to resubmit the manuscript of the work entitled, “Influence of the Computer-Aided Static Navigation Technique and Mixed Reality Technology on the Accuracy of the Orthodontic Micro-screws Placement. An in Vitro Study”

Reviewer 2: English language and style are fine/minor spell check required

Response: In order to adapt to the reviewer's 2 comments, we have send the manuscript to a specialized traductor.

Reviewer 2: The groups described in the material and methos section are different than the ones described in the abstract section. Please correct the abstract section describing that the groups included were from A to F.

Response: In order to adapt to the reviewer's 2 comments, we have rewrite the abstract section.

Reviewer 2: Is there any reference supporting the accuracy of the experimental models made by epoxy resin? If yes, please add as citation.

Response: In order to adapt to the reviewer's 2 comments, we clarify that unfortunately there is no reference supporting the accuracy of the experimental models.

Reviewer 2: There are a lot of tables and plots. Maybe it could be better to unify some tables or plots.

Response: In order to adapt to the reviewer's 2 comments, we have rewrite the Results section including a new study group, according to Reviewer 1 comments, trying to reduce the tables and plots.

Reviewer 2: Please correct the pagination from line 378.

Response: In order to adapt to the reviewer's 2 comments, we have corrected the pagination error.

We take this opportunity to thank the recommendations and suggestions made by the reviewers to improve the document.

Yours sincerely,